# Mono-Parasitic and Poly-Parasitic Intestinal Infections among Children Aged 36–45 Months in East Nusa Tenggara, Indonesia

**DOI:** 10.3390/tropicalmed8010045

**Published:** 2023-01-06

**Authors:** Alpha F. Athiyyah, Ingrid S. Surono, Reza G. Ranuh, Andy Darma, Sukmawati Basuki, Lynda Rossyanti, Subijanto M. Sudarmo, Koen Venema

**Affiliations:** 1Department of Child Health, Faculty of Medicine, Universitas Airlangga, Surabaya 60286, Indonesia; 2Department of Child Health, Dr. Soetmo General Hospital, Surabaya 60286, Indonesia; 3Food Technology Department, Faculty of Engineering, Bina Nusantara University, Jakarta 11480, Indonesia; 4Department of Medical Parasitology, Faculty of Medicine, Universitas Airlangga, Surabaya 60115, Indonesia; 5Malaria Study Group/Laboratory of Malaria, Institute of Tropical Disease, Universitas Airlangga, Surabaya 60286, Indonesia; 6Centre for Health Eating & Food Innovation, Maastricht University–Campus Venlo, 5928 SZ Venlo, The Netherlands

**Keywords:** intestinal parasitic infection, protozoa, helminth, young children, East Nusa Tenggara, Indonesia

## Abstract

The prevalence of intestinal parasitic infection remains high in developing countries, especially because of geographic and socio-demographic factors. This study aimed to evaluate intestinal parasitic infection, as well as its risk factors, among children aged 36–45 months in a rural area (North Kodi) and an urban area (Kupang) of East Nusa Tenggara, Indonesia. Anthropometry, socio-demographic factors and personal hygiene practices were assessed. A total of 214 children participated in the study, and 200 stool samples were collected for intestinal parasite examination. Approximately 30.5% (61/200) of the children were infected with one or more intestinal parasites (67.2%; 41/61 being mono-parasitic infections and 32.8%; 20/61 being poly-parasitic infections). A total of 85 intestinal parasites were detected, consisting of 35.3% (30/85) protozoa and 64.7% (55/85) helminths. The predominant protozoa were *Giardia lamblia* (43%; 13/30) and *Blastocystis* spp. (33.3%; 10/30), whereas the predominant helminths were *Trichuris trichiura* (50.9%; 28/55) and *Ascaris lumbricoides* (43.6%; 24/55). Moreover, intestinal parasitic infection was associated with rural area (OR 4.5; 95%CI 2.3–8.6); the absence of treatment with deworming drugs (OR 2.56; 95%CI 1.3–5.0); sanitation facilities without a septic tank (OR 4.3; 95%CI 2.1–8.5); unclean water as a source of drinking water (OR 4.67; 95%CI 2.4–9.4); no handwashing practice after defecation (OR 3.2; 95%CI 1.4–7.3); and stunted children (OR 4.4; 95%CI 2.3–8.3). In conclusion, poly-parasitic infections were common in this study. Poor personal hygiene practice and sanitation factors contributed to the high prevalence of intestinal parasitic infection in 36–45-month-old children in East Nusa Tenggara, Indonesia.

## 1. Introduction

Intestinal parasitic infections, particularly those of protozoa and helminth, are responsible for morbidity in children worldwide and represent a major public health problem in developing countries that is often neglected [1,2,3]. Children are particularly susceptible to infection by these microorganisms, which further negatively affects their nutritional status and physical development [1,2]. The global prevalence of intestinal parasitic infections remains high, with approximately 3.5 billion people infected and more than 200,000 deaths [4,5]. Based on data from 118 countries, the highest prevalence for soil-transmitted helminth infections (STHs) is detected in South Asia, Southeast Asia, and Sub-Saharan Africa. Of these infections, 67.3% occur in Asia [6]. In Indonesia, the prevalence of helminth infections is reported to vary between 2.5 and 62% [7]. East Nusa Tenggara, Indonesia, as a remote area, still experiences a high prevalence of parasitic infection. The STHs prevalence in Southwest Sumba and West Sumba in East Nusa Tenggara is >20%; however, the exact prevalence is difficult to ascertain. The prevalence of protozoan infections with *Entamoeba histolytica*, Giardia lamblia, and *Blastocystis hominis* (*Bh*) is reported to be 17.9% (76/424), 4.5% (19/424), and 34.4% (146/424), respectively [8].

In low- and middle-income countries, exposure to inadequate drinking water, sanitation, hygiene conditions, and hygiene behaviors are attributed to a greater proportion of intestinal parasite infections [9]. A recent systematic review reported that age, sex, residence, toilet facilities, washing hands with soap before a meal, shoe-wearing habits, trimming nails, eating undercooked food, personal hygiene and source of drinking water are the most risk factors for intestinal parasite infection [10]. Intestinal parasitic infections and their risk factors in Indonesia warrant further study. The current work was, therefore, carried out to evaluate the risk factors for intestinal parasitic infections among children aged 36–45 months in North Kodi and Kupang, East Nusa Tenggara, Indonesia.

## 2. Materials and Methods

### 2.1. Study Design and Ethical Clearance

A cross-sectional study was conducted on 214 children aged 36–45 months in Kupang and North Kodi, East Nusa Tenggara, Indonesia, from October to December 2021. The study protocol was approved by the Ethics Committee of the Research Institute of YARSI University, Jakarta, Indonesia, and registered at ClinicalTrials.gov with identifier number NCT05119218. The children’s parents or caregivers provided oral and written informed consent and signed a letter of consent before the children were included in the study.

### 2.2. Fecal Analysis and Parasitological Diagnosis

The parents/caregivers were instructed to collect each child’s stool and bring the stool specimen in a sterile container. All 200 collected stools were preserved in 10% formalin solution. The presence of parasites was detected via standard microscopy techniques (Olympus, Japan) using direct wet-mount smear methods, followed by staining with lugol solution, with six replications at the Laboratory of Malaria, Institute of Tropical Disease, Universitas Airlangga. The presence of eggs, larvae, trophozoites, or cysts was assessed for each type of helminth (*Trichuris trichiura*, Hookworm, *Ascaris lumbricoides*, and *Hymenolepis diminuta*) and protozoa (*Entamoeba histolytica*, *Entamoeba coli*, *Giardia lamblia*, and *Blastocystis*) spp., respectively. Positive parasitic infection was recorded by examining each prepared slide in which one or more parasites were detected. Mono-parasitic infection was defined as the presence of either one protozoan or one helminthic parasite in one child. Poly-parasitic infection was defined as (1) >1 positive protozoan parasites; (2) >1 helminthic parasites; and (3) mixed infection with both intestinal protozoan and helminthic parasites detected in one child.

### 2.3. Anthropometry

Height was measured using a wall stadiometer (Seca 208; precision, 0.1 cm) with the child’s head positioned according to the Frankfurt plane by trained nurses and general practitioners. Z-scores of height-for-age were calculated using the WHO AnthroPlus software provided by the World Health Organization, Geneva, Switzerland, in 2007 [11]. The height-for-age was classified as severely stunted (height for age < –3SD); stunted (–3SD to <–2SD); and normal (–2SD to +3SD). For analysis, the subjects were divided into the stunted group (if the subjects were severely stunted and stunted) and the normal group.

### 2.4. Subject Characteristics

A structured questionnaire was administered by two pediatricians, four general practitioners, nurses, laboratory technicians, and a community health care worker for face-to-face interviews with the respective child’s mother or caregiver, in order to collect sociodemographic information and hygiene practices. The independent variables were gender, locus of the rural (North Kodi) and urban (Kupang) area, mother’s education and occupation, family income, history of low birth weight, history of intake of deworming drugs in the last 6 months, family size, source of drinking water, type of sanitation facility, handwashing practice (before eating and after defecation), and handwashing facilities. Family size referred to the number of persons in the family and was categorized as small (<6 members), medium (6–8 members), and large (>8 members). Sanitation facility was defined as one that hygienically separates human feces from human contact and was categorized into a latrine with and without a septic tank. The mother’s education was divided into educated (elementary school, junior high school, senior high school, and university graduate) and uneducated. Family income was considered based on the regional minimum wage in each city and classified as lower than the regional minimum wage and equal to or greater than the regional minimum wage. The source of drinking water was divided into clean water (from mineral water, spring, tap water, and dug well) and unclear water (rainwater collection). Handwashing facilities were either fixed or mobile and included a sink with tap water and other models designated for handwashing.

### 2.5. Statistical Analyses

The data are presented as numbers and percentages for descriptive data. The chi-square and Fisher’s exact tests were used to assess differences in sociodemographic factors and hygiene practices between North Kodi and Kupang, as well as between stunted and normal children in categorical data. A univariate analysis was used to determine the odds ratio with the 95% confident interval of each variable that affected intestinal parasitic infection among children aged 36–45 months. Significance was set at *p* < 0.05. All statistical analyses were performed using the statistical program for social science (SPSS) Version 20.0 for Windows (SPSS Inc., Chicago, IL, USA).

## 3. Results

### Intestinal Parasitic Infection and Risk Factors in 36–45-Month-Old Children in East Nusa Tenggara, Indonesia

This study included 214 children, and 200 stool samples were collected to analyze the parasitic infection status and its risk factors. The flow chart of this study is shown in Figure 1. Children with incomplete stool sample data were excluded from the study. Appendix A show the demographic picture of Kupang and North Kodi. Kupang is a more densely populated urban area, whereas North Kodi is a rural area with mountains, hills and different types of housing compared with Kupang. A total of 30.5% (61/200) of the children were infected with intestinal parasites in this study, with 67.2% (41/61) of the cases being mono-parasitic infections and 32.8% (20/61) being poly-parasitic infections. Eighty-five intestinal parasites were detected in total, with 35.3% (30/85) being protozoan and 64.7% (55/85) being helminthic infections. *Giardia lamblia* (43%; 13/30) was the predominant protozoan parasite (Figure 2), whereas *Trichuris trichiura* (50.9%; 28/55) and *Ascaris lumbricoides* (43.6%; 24/55) were the predominant helminthic parasites (Figure 3) detected here.

Among children with mono-parasitic infection, 39% (16/41) had protozoan infection and 61% (25/41) had helminthic infection. *Giardia lamblia* and *Blastocystis* spp. were the most commonly detected parasites in protozoan mono-parasitic infection (50%; 8/16 and 25%; 4/16 of children, respectively). In turn, *Trichuris trichiura* and *Ascaris lumbricoides* were the most frequently detected parasites in children with helminthic mono-parasitic infection (44%; 11/25). In children with poly-parasitic infection, 10% (2/20) were infected with >1 protozoan, 50% (10/20) with >1 helminthic, and 40% (8/20) had mixed infections (both intestinal protozoan and helminthic parasites). Two children with >1 positive protozoa were detected to have *Entamoeba coli*–*Blastocystis* spp. and *Entamoeba coli–Giardia lamblia.* All children with >1 helminth infections were detected to have *Ascaris lumbricoides–Trichuris trichiura.* For mixed infection, two (25%) children were detected to have *Ascaris lumbricoides–Trichuris trichiura–Blastocystis* spp.; two (25%) with *Trichuris trichiura–Giardia lamblia–Blastocystis* spp.; two (25%) with *Trichuris trichiura–Giardia lamblia*; one (12.5%) with *Trichuris trichiura–Blastocystis* spp.; and one (12.5%) with *Ascaris lumbricoides–Entamoeba coli*. The microscopic findings of *Giardia lamblia*, *Entamoeba coli*, and *Blasocystis* spp. are depicted in Appendix A.

More protozoa were detected in children from Kupang (60%; 18/30), whereas more helminths were detected in children from North Kodi (98.2%; 54/55) (Figure 2 and Figure 3). *Entamoeba histolytica* (100%; 1/1), *Entamoeba coli* (83.3%; 5/6), and *Giardia lamblia* (61.5%; 8/13) were more frequent in Kupang children, whereas helminthic infection was exclusively caused by *Hymenolepis diminuta*. *Blastocystis* spp. (60%; 6/10) was the predominant pathogen among the protozoan infections recorded in North Kodi. All *Trichuris trichiura*, *Ascaris lumbricoides*, and Hookworm infections were detected in North Kodi (Figure 2 and Figure 3). The majority of stunted children had intestinal protozoan (93.3%; 28/30) and helminthic (69.1%; 38/55) infection. *Giardia lamblia* was the predominant intestinal protozoa in stunted children (42.9%; 12/30), followed by *Blastocystis* spp. (33.3%; 10/30) (Figure 4). *Trichuris trichiura* was the predominant intestinal helminth (38.2%; 21/55), followed by *Ascaris lumbricoides* (25.5%; 14/55) (Figure 5).

The univariate analysis revealed that living in a rural area, lack of treatment with deworming drugs, a latrine without a septic tank, unclean water as the source of drinking water, no handwashing practice after defecation, and stunted children were risk factors for intestinal parasitic infection in this study (Table 1). Significant differences in the education level of the mother, family income, deworming status, source of drinking water, type of sanitation facility, handwashing practice (before eating as well as after defecation), and handwashing facility (*p* < 0.05) were observed between children from Kupang and those from North Kodi. Other characteristics are described in Table 2. A history of low birth weight, no deworming, drinking unclean water, using a latrine without a septic tank, and no handwashing practice (before eating and after defecation) were significantly different in stunted children compared with normal children (*p* < 0.05) (Table 3).

## 4. Discussion

The most common protozoa detected in all stool samples was *Giardia lamblia*, followed by *Blastocystis* spp. This result is in line with a study reported by Maru et al. in Ethiopia; among 235 infected children, the most prevalent parasite was *Giardia lamblia* (20%) [12]. Moreover, a study performed by Diarthini in Karangasem (Bali, Indonesia) reported that the prevalence of *Blastocystis* spp. in elementary school children was 33% (35/103), which was similar to the present study [13]. Both of those studies were conducted under geographic and socio-hygiene conditions that were similar to those of East Nusa Tenggara, which could explain the similar results. The most common helminth detected in this study was *Ascaris lumbricoides*, followed by *Trichuris trichiura*. *Ascaris lumbricoides* and *Trichuris trichiura* (whipworm) are soil-transmitted helminths (STHs) that are mostly found in tropical and subtropical areas [14,15,16]. A study performed by Pullan et al. in 118 countries revealed that both *Ascaris lumbricoides* and *Trichuris trichiura* were the predominant global STHs infection agents [6]. Wani et al. reported a similar prevalence in India. Among 2256 children, the prevalence of *Ascaris lumbricoides* was the highest (68.3%), followed by *Trichuris trichiura* (27.9%) [17]. Higher *Trichuris trichiura* and *Ascaris lumbricoides* rates were also found in a study reported by Sungkar et al. in Sumba, Indonesia. Among 88 children, the rate of *Trichuris trichiura* was 85.2%, whereas that of *Ascaris lumbricoides* was 71.6% [18].

In addition to mono-parasitic infection, we also found poly-parasitic infection in this study. *Blastocystis* spp. was the predominant protozoa involved in poly-parasitic infections. This result was in line with the study performed by Diarthini et al. in Karangasem, Bali, Indonesia. *Blastocystis* spp. is a predominant parasite in under-developed countries, especially in children. *Blastocystis* spp. is often found in poly-parasitic infections with *Giardia lamblia*, *E. histolytica,* and *E. coli*. Moreover, it is detected in mixed infection with Hookworm [13]. In turn, *Ascaris lumbicoides* and *Trichuris trichiura* are the predominant helminths involved in poly-parasitic infections. In endemic areas, especially in warm tropical and sub-tropical areas, poly-parasitic infections occur frequently and might result in the exacerbation of morbidity, as well as a greater intensity of infection. Both helminths are transmitted via the faecal–oral route, which, because the exposure is similar, leads to a positive association. Poly-parasitic infections have no specific gastrointestinal symptoms; therefore, affected children are often underdiagnosed. If left untreated, moderate-to-heavy poly-parasitic infection could lead to chronic effects on the growth of children [14,19].

The total prevalence of intestinal parasitic infections in rural areas is notably higher than that recorded in urban areas. North Kodi is a rural area located in Southwest Sumba, whereas Kupang is the capital city, i.e., an urban area, and the administrative center of East Nusa Tenggara [20,21]. The hygiene and sanitation practices and facilities in urban and rural areas are different. The source of drinking water, type of sanitation facility, handwashing practice, and handwashing facility in North Kodi were significantly inferior compared with Kupang. Poor personal hygiene, poor environmental sanitation, low social economy, and population density will lead to an increase in soil-transmitted helminth and protozoa infections through the soil [22,23]. A greater number of children and their parents had no handwashing facility and used a latrine without a septic tank in North Kodi. In another study reported by Mane et al., students in a rural area reported a higher percentage (37.7%) of non-availability of a place for handwashing inside the home compared with only 17.9% in an urban area [24]. Based on the research of Idowu et al., a greater number of respondents (parents) from a rural community used an open pit. This was followed by the practice of open defecation by 46.7% of the responders. In contrast, the majority of the parents or caregivers from an urban area (49.1%) reported that they throw their children’s feces into a water closet after they use a potty [25].

Darlan et al. reported a strong relationship between personal hygiene practices and environmental sanitation and the incidence of soil-transmitted helminthic infection [26]. Another study from Apidechkul (North Thailand) performed among hill-tribe school children showed that drinking water contaminated by soil was an important risk factor for intestinal parasitic infection [27]. The practice of poor hygiene behaviors leads to a higher prevalence of soil contamination. The soil around rural areas is profoundly contaminated with parasite eggs stemming from the tendency to defecate without using a septic tank; soil-transmitted helminths require soil for immature stage development, in order to be transmitted to a host [28,29,30]. Unclean water as the source of drinking water remains a potential risk for helminthic infections not only because of direct ingestion, but also due to the consumption of unwashed fruits and vegetables or those that are improperly cooked [31].

Our study found that the deworming programs in the rural area were less frequent (55%) compared with the urban area. This was in line with a study reported by Sungkar et al. The rate of STH in 88 children in Sumba significantly decreased after deworming using Albendazole (from 95.4% to 53.4%) [18]. Deworming programs are routinely provided by the Indonesian Ministry of Health, with the distribution of Albendazole through primary healthcare or school-based deworming programs. Several lines of evidence suggest that these deworming drugs are used for public health intervention for preventing soil-transmitted helminth infection, with a lesser impact on protozoa. However, the outcome of this prevention strategy also depends on the environmental conditions and requires regular monitoring to observe the soil-transmitted helminths and the possibility of re-infection, which warrants the administration of an additional or adjusted dose of deworming drugs [32,33].

The lack of periodic evaluation of the nutritional status was one of the limitations of this study. Moreover, because some data were incomplete, the analyses were not performed using the same sample size. Finally, a PCR examination was not included as a diagnostic method for parasites; rather, wet staining and microscopic observation alone were used here. Conversely, the strength of this study was the employment of many types of variables, which allowed us to more precisely assess the risk of parasitic infection. The sample size was sufficient, and we compared samples from different geographic and social-hygiene conditions.

## 5. Conclusions

Mono-intestinal parasitic infections were most common in children aged 36-45 months in East Nusa Tenggara, Indonesia. Total intestinal protozoan infection was detected more in the urban area, Kupang, with *Giardia lamblia* and *Blastocystis* spp. being the predominantly detected pathogens. The total helminthic infection prevalence was higher in the rural area, North Kodi; the most-detected pathogen was *Ascaris lumbricoides*, followed by *Trichuris trichiura*. Poly-parasitic intestinal infection was observed in more than one-third of infected children. Living in a rural area, lack of treatment with deworming drugs, use of a latrine without a septic tank, unclean water as the source of drinking water, no handwashing practice after defecation, and stunted children were the risk factors for intestinal parasitic infection observed in this study. We urge routine deworming every 6 months, providing clean water by building a drill well, and educating the community with good personal hygiene practice. Further study is currently ongoing to prove the efficacy of these comprehensive and holistic treatments in eradicating the parasitic infection. The scientific evidence obtained in this study will be recommended to local government as public policy.

## Figures and Tables

**Figure 1 tropicalmed-08-00045-f001:**
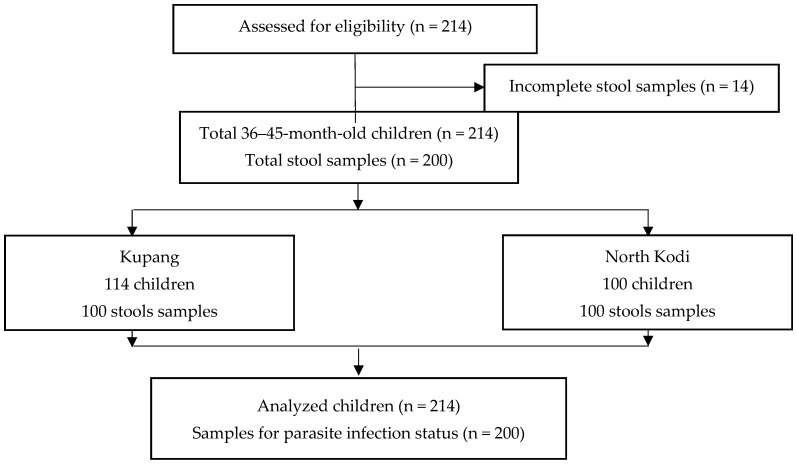
Flow diagram of this study.

**Figure 2 tropicalmed-08-00045-f002:**
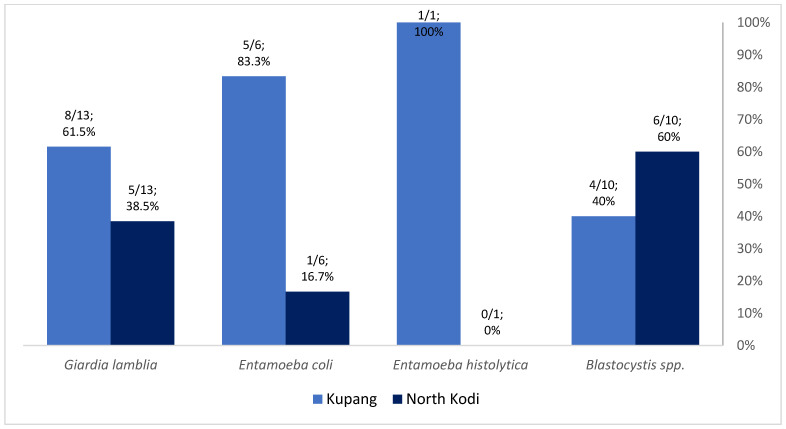
Intestinal protozoan distribution in Kupang and North Kodi, East Nusa Tenggara; n = 30 children. *Giardia lamblia* was detected in a total of 13 children; *Entamoeba coli* was detected in a total of 6 children; *Entamoeba histolytica* was detected in a total of 1 child; *Blastocystis* spp. was detected in a total of 10 children (one child had ≥1 protozoan parasites).

**Figure 3 tropicalmed-08-00045-f003:**
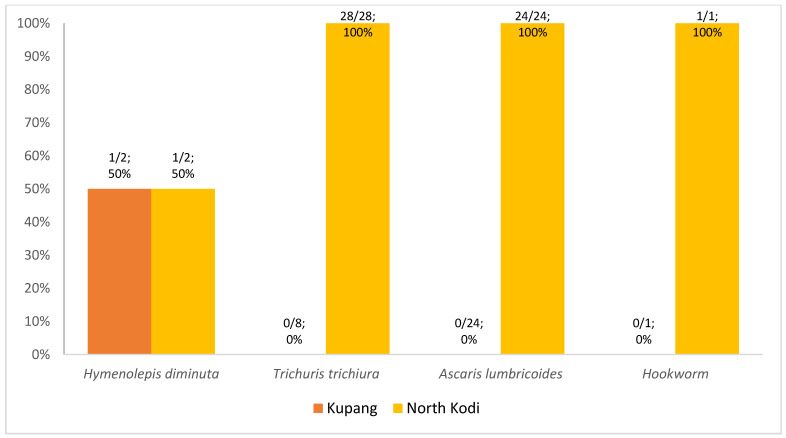
Intestinal helminthic distribution in Kupang and North Kodi, East Nusa Tenggara; n = 51 children. *Hymenolepis diminuta* was detected in a total of 2 children; *Trichuris trichiura* was detected in a total of 28 children; *Ascaris lumbricoides* was detected in a total of 24 children; *Hookworm* was detected in a total of 1 child (one child had ≥1 helminthic parasites).

**Figure 4 tropicalmed-08-00045-f004:**
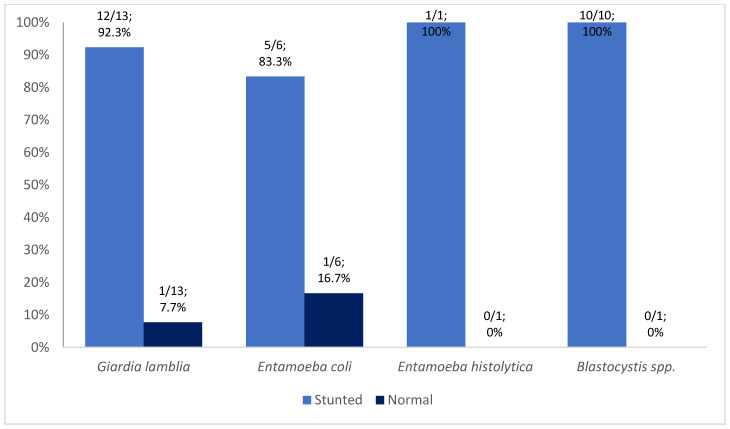
Intestinal protozoan distribution in stunted and normal children in East Nusa Tenggara; n = 30 children. *Giardia lamblia* was detected in a total of 13 children; *Entamoeba coli* was detected in a total of 6 children; *Entamoeba histolytica* was detected in a total of 1 child; *Blastocystis* spp. was detected in a total of 10 children (one child had ≥1 protozoan parasites).

**Figure 5 tropicalmed-08-00045-f005:**
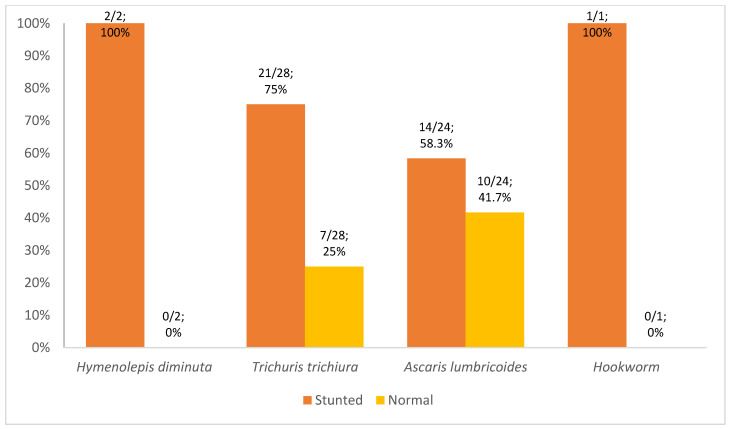
Type of helminthic in stunted and normal children in East Nusa Tenggara; n = 51 children. *Hymenolepis diminuta* was detected in a total of 2 children; *Trichuris trichiura* was detected in a total of 28 children; *Ascaris lumbricoides* was detected in a total of 24 children; *Hookworm* was detected in a total of 1 child (one child had ≥1 helminthic parasites).

**Table 1 tropicalmed-08-00045-t001:** Univariate analysis of intestinal parasitic infection risk factors among children aged 36–45 months in Kupang and North Kodi, East Nusa Tenggara Timur, Indonesia.

Risk Factors	Intestinal Parasitic Infection		*p* Value *	OR (95% CI)
	Infected n (%)	Not Infected n (%)		
Gender			0.203	1.5 (0.8–2.8)
Boy	33 (35.5)	60 (64.5)		
Girl	28 (26.2)	79 (73.8)		
Locus			0.0001	3.8 (2.0–7.4)
Rural area	44 (44)	56 (56)		
Urban area	17 (17)	83 (83)		
Family size			0.171	2.1 (0.8–5.1)
>8 members	10 (45.5)	12 (54.5)		
≤8 members	51 (28.7)	127 (71.3)		
Mother’s education (n = 198)			0.094	2.1 (1.0–4.6)
Uneducated	14 (45.2)	17 (54.8)		
Educated	47 (28.1	120 (71.9)		
Mother’s occupation (n = 194)			0.609	1.3 (0.6–2.8)
Unemployed	47 (31.1)	104 (68.9)		
Employed	11 (25.6)	32 (74.4)		
Family income (n = 192)			0.692	1.3 (0.6–2.7)
<Regional minimum wage	45 (30.2)	104 (69.8)		
≥Regional minimum wage	11 (25.6)	32 (74.4)		
Low birth weight (n = 181)			0.406	1.6 (0.7–3.5)
Yes	11 (36.7)	19 (63.3)		
No	41 (27.2)	110 (72.8)		
Deworming			0.020	2.3 (1.2–4.4)
No	23 (44.2)	29 (55.8)		
Yes	38 (25.7)	110 (74.3)		
Type of sanitation facility			0.0001	4.3 (2.0–9.2)
Latrine without septic tank	21 (58.3)	15 (41.7)		
Latrine with septic tank	40 (24.4)	124 (75.6)		
Source of drinking water			0.0001	3.89 (2.0–7.8)
Unclean water	24 (54.5)	20 (45.5)		
Clean water	37 (23.7)	119 (76.3)		
Handwashing before eating			0.671	1.3 (0.6–3.1)
No	10 (35.7)	18 (64.3)		
Yes	51 (29.7)	121 (70.3)		
Handwashing after defecation			0.018	2.9 (1.3–6.6)
No	14 (51.9)	13 (48.1)		
Yes	47 (27.2)	126 (72.8)		
Stunted growth (n = 199)			0.0001	3.4 (1.8–6.6)
Yes	43 (43.0)	57 (57)		
No	18 (18)	82 (82)		
Handwashing facility			0.194	3.0 (0.8–11.6)
No	5 (55.6)	4 (44.4)		
Yes	56 (29.3)	135 (70.7)		
Total	100 (50)	100 (50)		

* Chi-squared test, significance set at *p* < 0.05.

**Table 2 tropicalmed-08-00045-t002:** Sociodemographic and hygiene practice characteristics of children aged 36–45 months from Kupang and North Kodi, East Nusa Tenggara Timur, Indonesia.

Variables	Kupang n (%)	North Kodi n (%)	*p* Value *
Gender			0.442
Boy	51 (50.5)	50 (50)	
Girl	63 (55.8)	50 (50)	
Family size			0.473
Small (<6 members)	71 (62.3)	55 (55)	
Medium (6–8 members)	32 (28.1)	31 (31)	
Large (>8 members)	11 (9.6)	14 (14)	
Mother’s education			0.0001
Uneducated	4 (3.5)	28 (28)	
Educated	110 (96.5)	72 (72)	
Mother’s occupation (n = 208)			0.071
Unemployed	91 (56.9)	69 (43.1)	
Employed	20 (41.7)	28 (58.3)	
Family income (n = 206)			0.044
<Regional minimum wage	93 (58.5)	66 (41.5)	
≥Regional minimum wage	19 (40.4)	28 (59.6)	
Low birth weight (n = 195)			1.000
Yes	18 (58.1)	13 (41.9)	
No	96 (58.5)	68 (41.5)	
Deworming			0.0001
Yes	107 (93.9)	55 (55)	
No	7 (6.1)	45 (45)	
Source of drinking water			0.0001
Clean water	114 (100)	54 (54)	
Unclean water	0 (0)	46 (46)	
Type of sanitation facility			0.0001
Latrine with septic tank	77 (67.5)	18 (18)	
Latrine without septic tank	37 (32.5)	82 (82)	
Handwashing practice (before eating)			0.0001
Yes	110 (96.5)	76 (76)	
No	4 (3.5)	24 (24)	
Handwashing practice (after defecation)			0.0001
Yes	112 (98.2)	75 (75)	
No	2 (1.8)	25 (25)	
Handwashing facility			0.014
Yes	113 (99.1)	92 (92)	
No	1 (0.9)	8 (88.9)	
Height for age status			0.187
Stunted	49 (43)	52 (52)	
Normal	65 (57)	48 (48)	
Total	114 (53.3)	100 (46.7)	

* Chi-squared test, significance set at *p* < 0.05.

**Table 3 tropicalmed-08-00045-t003:** Socio-demographic and hygiene practice characteristics of stunted and normal children aged 36–45 months in East Nusa Tenggara Timur, Indonesia.

Variables	Stunted n (%)	Normal n (%)	*p* Value *
Gender			0.927
Boy	48 (47.5)	53 (46.9)	
Girl	53 (52.5)	60 (53.1)	
Family size			0.449
Small (<6 members)	63 (62.4)	63 (55.8)	
Medium (6–8 members)	18 (21.8)	28 (36.3)	
Large (>8 members)	20 (15.8)	22 (8)	
Mother’s education level			0.730
Uneducated	16 (15.8)	16 (14.2)	
Educated	85 (84.2)	97 (85.8)	
Mother’s occupation (n = 208)			0.909
Unemployed	77 (48.1)	83 (51.9)	
Employed	22 (45.8)	26 (54.2)	
Family income (n = 206)			0.834
<Regional minimum wage	76 (47.8)	83 (52.2)	
≥Regional minimum wage	21 (44.7)	26 (55.3)	
Low birth weight (n = 195)			0.015
Yes	21 (67.7)	10 (32.3)	
No	69 (42.1)	95 (57.9)	
Deworming			0.017
Yes		93 (82.3)	
No	32 (31.7)	20 (17.7)	
Source of drinking water			0.036
Clean water	73 (72.3)	95 (84.1)	
Unclean water	28 (27.7)	18 (15.9)	
Type of sanitation facility			0.0001
Latrine with septic tank	30 (29.7)	65 (57.5)	
Latrine without septic tank	71 (70.3)	48 (42.5)	
Handwashing practice (before eating)			0.019
Yes	82 (81.2)	104 (92)	
No	19 (18.8)	9 (8)	
Handwashing practice (after defecation)			0.030
Yes	83 (82.2)	104 (92)	
No	18 (17.8)	9 (8)	
Handwashing facility			0.060
Yes	94 (45.9)	111 (54.1)	
No	7 (77.8)	2 (22.2)	
Locus			0.187
Urban area	49 (43)	65 (57)	
Rural area	52 (52)	48 (48)	
Total	114 (53.3)	100 (46.7)	

* Chi-squared test, significance was set at *p* < 0.05.

## Data Availability

Not applicable.

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
