# Peer review of "Mono-Parasitic and Poly-Parasitic Intestinal Infections among Children Aged 36–45 Months in East Nusa Tenggara, Indonesia"

_tropicalmed, 2023, doi:10.3390/tropicalmed8010045_

Round 1

Reviewer 1 Report

I noticed a minor grammar issue on Line 62 "current work 'as" therefore carried out" should be was?

Figures 4 & 5 are hard to read on their own, I highly recommend adding titles and labels to guide the reader into understanding why the denominators in the subgroups do not add up to 100 and why the denominators change across categories. If the title or a legend can explain the denominators, it will clarify both figures.

Final conclusions, based on these results do you have a recommendation for further research, policy change, public health practice or programming? After such a rigorous analysis, the conclusion falls a bit flat. What can you say based on these results and what would you recommend for further research? Can you draw conclusions from this that relate to directions in public health policy or practice? 

Author Response

  1. Issue on line 62 already revised to “current work was”
  2. We simplify Figure 4 (line 240 – 243) and 5 (line 247 – 250) to make it easier to read for the reader as attached. And for the figure 1 (line 211 – 214) and 2 (line 216 – 219) as well.
  3. The final conclusion have been revised according to input from referee (line 337 – 350)

Reviewer 2 Report

The article belongs to important primary research, based on which measures can be taken to improve parasitic infections.

Title

2 Mono-paraisitic – should be parasitic

Abstract

26 Blastocystis (33.3%; 10/30), is it Blastocystis hominis? – should be corrected in the whole manuscript, either state the abbreviation while first mentioning each species, e. g. Blastocystis hominis (B.hominis, or BH) and use these abbreviations further in the text, or always state the full name of the parasite.

53 Entamoeba histolytic - should be histolytica

54 Blastocystis hominis (Bh) - abbreviation (Bh) is not necessary if it’s not used further in the text.

Methods

126 All statistical analyses were performed using SPSS Version 20.0 for Windows.  – please explain the abbreviation and specify the manufacturer.

Results

140, 145, 162 - Giardia lambia  - should be lamblia

162 - histolitica (100%; 1/1), histolytica

164, Figure 3 - Hymenolepis diinutashould be diminuta

Author Response

  1. The typo mention already corrected (line 2, line 53, line 143, line 148, line 165, line 167)
  2. The full name of the parasite should be Blastocystis spp. and alreacy corrected in the manuscript (line 25, line 84, line 148, line 155, line 158 – 160, line 162, line 167, line 173, line 254, line 257, line 272, line 275 – 276)
  3. The abbreviation for SPPS already explained “statistical program for social science (SPSS)” and the manufactured already written in the manuscript (line 128 – 129)